# A Streamlined Approach for Fluorescence Labelling of Low-Copy-Number Plasmids for Determination of Conjugation Frequency by Flow Cytometry

**DOI:** 10.3390/microorganisms11040878

**Published:** 2023-03-29

**Authors:** Qin Qi, Muhammad Kamruzzaman, Jonathan R. Iredell

**Affiliations:** 1Centre for Infectious Diseases and Microbiology, The Westmead Institute for Medical Research, The University of Sydney, Westmead, Sydney, NSW 2145, Australia; 2Westmead Hospital, Westmead, Sydney, NSW 2145, Australia

**Keywords:** plasmid, conjugation, fluorescence labelling, flow cytometry, horizontal gene transfer

## Abstract

Bacterial conjugation plays a major role in the dissemination of antibiotic resistance and virulence traits through horizontal transfer of plasmids. Robust measurement of conjugation frequency of plasmids between bacterial strains and species is therefore important for understanding the transfer dynamics and epidemiology of conjugative plasmids. In this study, we present a streamlined experimental approach for fluorescence labelling of low-copy-number conjugative plasmids that allows plasmid transfer frequency during filter mating to be measured by flow cytometry. A blue fluorescent protein gene is inserted into a conjugative plasmid of interest using a simple homologous recombineering procedure. A small non-conjugative plasmid, which carries a red fluorescent protein gene with a toxin–antitoxin system that functions as a plasmid stability module, is used to label the recipient bacterial strain. This offers the dual advantage of circumventing chromosomal modifications of recipient strains and ensuring that the red fluorescent protein gene-bearing plasmid can be stably maintained in recipient cells in an antibiotic-free environment during conjugation. A strong constitutive promoter allows the two fluorescent protein genes to be strongly and constitutively expressed from the plasmids, thus allowing flow cytometers to clearly distinguish between donor, recipient, and transconjugant populations in a conjugation mix for monitoring conjugation frequencies more precisely over time.

## 1. Introduction

Bacterial conjugation is an important example of horizontal gene transfer. It is a major source of genetic innovation for bacteria and a driving force for the rapid evolution of bacterial genomes [1,2,3]. Conjugative plasmids generally carry the genes of the conjugation machinery that are required for the unidirectional cell-to-cell transfer of plasmid DNA [4,5], but regulatory mechanisms for the conjugation machinery often vary according to plasmid incompatibility groups [6]. During conjugation, plasmids are transferred from donor cells (D) to recipient cells (R) to form transconjugants (T), which are recipients that have taken up conjugative plasmids. Due to the important roles that conjugation plays in spreading antimicrobial resistance (AMR), virulence, and virulence-associated genes between bacterial pathogens, there have been calls for more robust and standardized quantitative methods for conjugation frequency measurements [7,8,9,10].

To date, there have been numerous experimental methods for quantifying frequencies at which plasmids are transferred by conjugation (Table 1) [7,10]. Regardless of the experimental approach, all these methods generally involve co-incubating a donor (D) with a recipient (R) bacterial population under specific in vitro or in vivo experimental conditions for a specified duration. In classical experimental approaches, transconjugants (T) are isolated on antibiotic-containing selective media that eliminate donors and recipients. Determining the densities of D, R, and T requires serial dilution of bacteria populations and plating on appropriate antibiotic-containing selective media before and after conjugation. The densities are subsequently determined by counting the number of colony-forming units (CFU) after a specified duration of incubation. Major drawbacks of classic plating and CFU counting approaches include their low precision [11,12,13,14,15] and labor-intensive procedures. Moreover, using antibiotics for selecting transconjugants has the potential to alter conjugation frequency through the effects of antibiotics on the conjugation machinery encoded by plasmids of interest [16,17].

Flow cytometry is a powerful experimental technique that offers an attractive alternative to the plating and CFU counting method [7]. To date, several studies have demonstrated the feasibility of a fluorescence-labelling approach that uses two different fluorescent protein genes. Conjugative plasmids are often labelled with green fluorescent protein genes [7,26,27,28,29,30], while the chromosomes of donor strains (D) [7,27,29,30] are tagged with a red fluorescent protein gene such as mCherry. This allows the three sub-populations D, R, and T to be uniquely identified in a conjugation mix based on differences in their fluorescence emission spectra. When randomly sampled individual cells in a conjugation mix are excited by lasers in a flow cytometer, D emits both green and red fluorescence (*gfp*^+^ mCherry^+^), while T emits positive signals for *gfp* only. R cells are detected as non-fluorescent cells. An alternative approach is to label the chromosome of recipients (R) with mCherry [28]. In this case, T becomes the double-positive (*gfp*^+^ mCherry^+^) population. Another approach that circumvents chromosomal tagging of bacterial strains is to label recipient strains with a small mCherry-bearing, non-conjugative plasmid. This method is more timesaving than chromosomal tagging procedures for generating mCherry-bearing recipient strains and has been successfully applied in flow cytometry-based filter mating assays [7,31]. However, small non-conjugative plasmids are usually unstable without antibiotic selection for that plasmid. Therefore, fluorescent-labelled small non-conjugative plasmids can be lost from the recipient bacterial population during antibiotic-free conjugation experiments, thus affecting accurate measurement of transconjugant frequencies.

In this study, we present a two-color fluorescence-labelling approach for quantifying the transfer rate of a low-copy-number conjugative plasmid during filter mating. We designed a non-conjugative, low–medium-copy-number plasmid that not only strongly expresses mCherry (Figure 1), but also carries a toxin–antitoxin (TA) system from an *E. coli* IncF plasmid [32,33,34]. This TA system functions as an “addiction” module which ensures that the mCherry plasmid is stably maintained even in the absence of antibiotic selection pressure. We also describe a simple homologous recombineering technique that can be applied to fluorescent labelling of low-copy-number conjugative plasmids of interest. Fluorescent protein-encoding genes are strongly expressed from a strong constitutive promoter, which enhances the ability of plasmid-bearing cells to produce strong fluorescence signals to be reliably detected by flow cytometers. Rigorous quality control and verification procedures were incorporated to improve the reliability of our experimental workflow. It is hoped that the methods and template plasmids that we developed in this work will be useful for the plasmid research community.

## 2. Materials and Methods

### 2.1. Bacterial Strains

Rifampicin-resistant mutants of *E. coli* UB5201Rf reference strain were used as the donor strain of the *ebfp2* (blue fluorescent protein gene)-tagged IncM curing plasmid pJIMK46 (Table 2). The amino acid substitution in the gene product of *rpoB* that confers rifampicin resistance in UB5201Rf was found to be S512P based on Sanger sequencing of the rifampicin resistance-determining region (Australian Genome Research Facility, Sydney, Australia) [35]. The sodium azide-resistant *E. coli* J53Az reference strain was used as the recipient strain [36,37]. The characteristics of the bacterial strains, primers, and plasmids used in this study are summarized in Table 2, Table 3 and Table 4, respectively.

To construct the p_mCherry-stable (pBAD33::P_c_S-mCherry-*ccdAB*) plasmid for mCherry red fluorescence-tagging of J53Az, the mCherry gene was PCR amplified from the pNEO-mCherryht plasmid template [41] using the primers mCherry_*Xba*I-F and mCherry_*Sac*I-R. The toxin–antitoxin system *ccdAB* from an *E. coli* IncF plasmid (pJIE134) was PCR amplified with the primers *ccdAB*_*EC_Hpa*I-F and *ccdAB*_*EC*_*Sac*I-R [32]. mCherry and *ccdAB* were sub-cloned into the pSS9 vector at the *Xba*I/*Sac*I and *Sac*I/*Hpa*I sites, respectively, to create pSS9_mCherry [39]. The combined mCherry-*ccdAB* DNA fragment was cloned into the pBAD33-Gm vector at the *Xba*I/*Hpa*I sites [40]. A strong constitutive promoter sequence P_C_S from a class 1 integron with the ribosomal binding site 5′-AGGAGG-3′ was PCR amplified from the pJIJP005 plasmid [42] template using the primers P_c_S_*Xba*I-F and P_c_S_*Xba*I-RBS-R and cloned upstream of mCherry-*ccdAB* at the *Xba*I site. Chemically competent J53Az cells were transformed with the resulting p_mCherry-stable plasmid by heat shock and selected on LB agar containing gentamicin (8 µg/mL).

To tag pJIMK46 with *ebfp2*, an oligonucleotide fragment P_C_S-*ebfp2* with codon optimization for the *ebfp2* coding region (IDT, Coralville, IA, USA) was TA-cloned into the NEB pMiniT vector (New England Biolabs, Ipswich, MA, USA) to generate the pMini_blue-strong (pMiniT-P_C_S-*ebfp2*) template plasmid. The P_C_S-*ebfp2* fragment was PCR amplified using the primer pair pJIMK46::P_c_S-*ebfp2*-F and -R, which introduced 40 bp homologous flanking regions for integration between the *trbC* and *tetA* genes in pJIMK46 (Figure 2). The PCR product was electroporated into electrocompetent cells of J53Az + pJIMK46 + pKM200. Colonies of J53Az + pJIMK46::P_C_S-*ebfp2* were isolated on LB agar containing 50 µg/mL ampicillin. The resulting pConj_blue-strong (pJIMK46::PCS-ebfp2) plasmid was transferred from J53Az into *E. coli* UB5201Rf by conjugation. GenBank annotations of key plasmids generated in this work can be accessed from the Dryad Data Repository (refer to Data Availability Statement).

### 2.2. Bacterial Growth Rate Assays

Growth rate assays for the *E. coli* J53Az control strain and the mCherry-tagged recipient strain (J53Az + p_mCherry-stable) were performed in 4 biological replicates on 4 different days. Stationary phase cultures were adjusted to an optical density (OD_600_) of approximately 1.0 in antibiotic-free LB broth. The cultures were further diluted 1:250 in antibiotic-free LB broth and transferred as 200 µL cultures into the wells of a 96-well microplate (Corning, New York, USA) in 3–4 technical replicates. The microplate was incubated overnight with shaking at 37 °C in a SpectraMax iD5 Multi-Mode Microplate Reader (Molecular Devices, San Jose, CA, USA) during which OD_600_ was measured every 10 min. The OD_600_ readings during the exponential phase were blank corrected and log transformed. An array of linear regression values within the 0.02 < OD_600_ < 0.18 interval was calculated for each culture using every combination of six consecutive data points available for each culture. A sliding window was used to determine the maximum exponential growth rate within this interval for the 4 biological replicates. Welch’s *t*-test was applied to compare the average final blank corrected OD_600_ readings reached by stationary phase cultures of the two strains after 15 h of growth for the 4 biological replicates.

### 2.3. Flow Cytometry-Based Measurement of Transconjugant Frequency

Filter mating was performed in 4 biological replicates on 4 different days. Stationary phase cultures of the recipient strain J53Az + p_mCherry-stable and the donor strain UB5201Rf + pConj_blue-strong were grown in LB containing 8 µg/mL gentamicin and 50 µg/mL ampicillin, respectively at 37 °C with shaking. Each culture was washed in antibiotic-free LB broth and adjusted to OD_600_ = 0.9 ± 0.1. Next, 1.6 mL donor and recipient strain cultures were mixed, pelleted, and re-suspended in 170 µL LB medium. Concentrated cultures of the single-color donor and recipient strains were prepared in an identical way. We transferred 40 µL drops of each cell suspension onto individual Whatman cellulose nitrate membranes (GE Healthcare, Chicago, IL, USA) placed on antibiotic-free 1.5% LB agar plates. At the start of the conjugation experiment, one membrane containing each cell suspension was re-suspended in 4 mL sterile-filtered PBS containing 0.2 mM EDTA and vortexed to dislodge bacterial cells. This procedure was repeated for the remaining membranes every 2 h of incubation on LB agar plates at 37 °C up to 6 h. The recovered cell suspensions were further diluted 1:250 in 1 mL PBS + 0.2 mM EDTA. A culture of the non-fluorescent J53Az strain was also diluted at 1:1000 in PBS + 0.2 mM EDTA to be used as the non-fluorescent control strain. 

The diluted bacterial suspensions were well vortexed prior to analysis on the BD FACSymphony flow cytometer (BD Biosciences, Franklin Lakes, NJ, USA). The mCherry fluorophore was excited by the yellow laser and detected through a 610/20 nm bandpass filter. The *ebfp2* fluorophore was excited by the violet laser and detected through a 474/25 nm bandpass filter. Fluorescent beads with 0.88 µm and 1.34 µm diameters from the Size Standard Kit (Spherotech, Lake Forest, IL, USA) were used to validate the performance of the flow cytometer in detecting particle sizes. For each sample, 10^5^ events were acquired and recorded by the flow cytometer within an upper time limit of 10 min. 

### 2.4. Verification of Transconjugant Frequency by Selective Agar Plate Count

Following the same filter mating protocol as the flow cytometry-based workflow, transconjugant frequency of the fluorescence labelled plasmid pConj_blue-strong was measured using the conventional selective agar plate count method [7]. Conjugation mixtures at 2 h, 4 h, and 6 h time points were suspended in 4 mL sterile-filtered PBS containing 0.2 mM EDTA and vortexed to dislodge bacterial cells. The cell suspension was serially diluted with sterile-filtered PBS and spread plated on to LB agar plates containing sodium azide (100 µg/mL) or sodium azide (100 µg/mL) + ampicillin (75 µg/mL). Sodium azide + ampicillin and sodium azide-only plate counts provided the transconjugants (T) and the T + R estimates, respectively. Transconjugant frequency was calculated by dividing T by T + R. 

### 2.5. Flow Cytometry Data Analysis

Flow cytometry data analysis was performed using the FlowJo software (v10.7, FlowJo LLC, Ashland, OR, USA). Briefly, a universal rectangular gate was used to separate bacterial cells from background noise events in all SSC-A vs. FSC-A plots (Panels A and B, Appendix A, Appendix A). The autogating tool was used to capture approximately 90% of all events based on the contour of each bivariate plot within each rectangular gate (Panel C, Appendix A, Appendix A). Doublet discrimination was performed on SSC-H vs. SSC-A bivariate plots by drawing a narrow rectangular gate along the diagonal to retain single cells (Panel D, Appendix A, Appendix A). Compensation matrices were calculated for single-color control strains for mCherry^+^ and *ebfp2*^+^ from the 4 h time point and were applied to all samples. A spider gate was drawn on the compensated mCherry vs. compensated BFP bivariate plots to distinguish between J53Az (double negative), UB5201Rf + pConj_blue-strong control (*ebfp2*^+^) and J53Az + p_mCherry-stable control (mCherry^+^) populations. 

## 3. Results and Discussion

Here, we describe an application of our fluorescent-labelling method for quantifying the proportions of donors (D), recipients (R), and transconjugants (T) in a conjugation mix during filter mating. Time course conjugation assays were performed to obtain the transconjugant frequencies T/(T + R) of a blue fluorescent protein gene (*ebfp2*)-tagged, low-copy-number IncM plasmid, pJIMK46, in a conjugation mix of *E. coli* UB5201Rf donor and mCherry-labelled J53Az recipient strains (Table 2).

### 3.1. Labelling of E. coli J53Az Recipient Strain with a mCherry-Bearing Non-Conjugative Small Plasmid

The recipient strain was the sodium azide-resistant *E. coli* J53Az strain containing a small mCherry-bearing, non-conjugative plasmid p_mCherry-stable. This plasmid has a low–medium-copy-number p15A replicon [40]. A strong constitutive integron promoter (P_C_S) from a class 1 integron in the multi-drug resistance (MDR) region of the IncM plasmid pJIBE401 was cloned upstream of mCherry [43]. Gene expression from P_C_S (cloned from a class 1 integron) is approximately five times stronger than the constitutive P_TAC_ promoter [42]. Using a strong promoter ensures that recipient cells produce sufficiently strong red fluorescence signals to be distinguishable from the non-fluorescent J53Az negative control strain despite being expressed from a low–medium-copy-number plasmid. 

To test the strength and stability of mCherry signals from the recipient strain in an antibiotic-free environment, a stationary phase culture of J53Az + p_mCherry-stable was concentrated in LB medium, transferred to two pieces of Whatman membranes placed on LB agar without antibiotics, and incubated at 37 °C under static conditions. The recovered bacterial cells, both from the start (time = 0 h; Figure 1B) and the end of the incubation (time = 6 h; Figure 1C), can be clearly distinguished from the J53Az negative control (Figure 1A) when mCherry signals from these populations were quantified by a flow cytometer. This suggests that sufficiently strong mCherry fluorescence signals were produced for the recipients to be fully distinguishable from mCherry-negative cells, and that the plasmid can be stably maintained in recipient cells without antibiotics. The stability of fluorescent proteins such as mCherry is affected by environmental factors such as oxygen availability, pH, and choice of growth media, all of which can affect the correct maturation of fluorescent proteins [27]. It is therefore essential that single-color controls such as those shown in Figure 1 are included throughout the entire time course of conjugation assays.

To verify that the high expression of mCherry did not impose a fitness burden on the J53Az host cells, we compared the exponential growth rates of J53Az and J53Az + p_mCherry-stable in antibiotic-free LB medium at 37 °C. The maximum exponential growth rate of J53Az + p_mCherry-stable strain relative to the J53Az strain was 1.01 ± 0.01 (mean ± standard error of the mean; *n* = 4). There were also no significant differences in the final cell densities in the stationary phase reached by both cultures after 15 h of growth—the mean final blank corrected OD_600_ for J53Az and J53Az + p_mCherry-stable cultures were 1.00 ± 0.03 and 1.04 ± 0.01, respectively (Welch’s two-sample *t*-test: *t* = −1.23, d.f = 4.26, *p* = 0.28). These growth rate data suggest that mCherry expression did not negatively affect the growth characteristics of the J53Az host strain in antibiotic-free LB medium.

Conjugation frequencies are not only affected by conjugation-related genes in conjugative plasmids themselves. Recent in vitro and in vivo studies showed that the conjugation frequencies of the same plasmids varied significantly when different combinations of donor and recipient strains were paired, which suggests that the genotypic and phenotypic traits of different donor and recipient strains can also affect conjugation frequency [18,44,45,46]. This underscores the necessity for more combinations of donor and recipient strains to be tested when measuring conjugation frequencies. The p_mCherry-stable plasmid can facilitate rapid mCherry-labelling of candidate recipient strains by heat shock or electroporation. These procedures are faster than chromosomal labelling techniques such as CRISPR gene editing and transposon mutagenesis. However, if the conjugative plasmids of interest contain *ccdAB*-type TA modules, the *ccdAB* operon in p_mCherry-stable should then be replaced with unrelated type II TA modules, such as *pemIK*, *parDE*, or *relBE*, to prevent cross-talk between two pairs of similar addiction modules.

### 3.2. Labelling of Low-Copy-Number Plasmids with Blue Fluorescent Protein Gene

pJIMK46 is an IncM plasmid (Figure 2A) that was engineered from the pJIBE401 plasmid of a clinical origin by deleting its entire MDR region and its *pemK* toxin gene [24]. In a previous work, this interference plasmid (also known as curing plasmid) was developed to displace and cure other AMR IncM plasmids from resident *E. coli* hosts in the mouse gut in vivo. We used pJIMK46 as an example of a low-copy-number conjugative plasmid with a low number of antibiotic resistance genes, for which a simple one-step homologous recombineering procedure can be adopted for fluorescence tagging. A PCR product containing a blue fluorescent protein gene (*ebfp2*) downstream of P_C_S promoter, as well as an ampicillin-resistance gene *bla*, was first generated (Figure 2B). Using 60 bp primers, we introduced 40 bp overhanging bases that are homologous to the integration sites in pJIMK46 (Table 3). A lambda-red recombinase expressing *E. coli* host strain was electroporated with the PCR product, allowing the P_C_S-*ebfp2-bla* fragment to be inserted between *tetA* and *trbC* in pJIMK46 by homologous recombineering. The *ebfp2-*tagged pJIMK46 was transferred into the *E. coli* UB5201Rf host strain. The strength of blue fluorescence signals from these two strains was sufficiently strong to distinguish them from the non-fluorescent J53Az negative control on a flow cytometer despite the low-copy number of the IncM plasmid backbone. The blue fluorescence signals were also stable in the absence of antibiotic selection pressure (Figure 3).

### 3.3. Quantifying Transconjugant Frequencies Using Flow Cytometry and Conventional Plate Count Methods

Filter mating was performed using early stationary phase cultures of the J53Az + p_mCherry-stable recipient strain (R) and UB5201Rf + pConj_blue-strong donor strain (D) that were grown in antibiotic-free LB medium. The donor and recipient strains were mixed in 1:1 ratio at the start of the mating experiment. The conjugation mix was evenly distributed onto four Whatman filter membranes placed on 1.5% LB agar, which were incubated at 37 °C for 6 h. At the start of the experiment and every 2 h during conjugation, bacterial cells were recovered from the conjugation mix by vortexing one filter membrane per time point in sterile PBS with 0.2 mM EDTA. At low concentrations, EDTA destabilizes biofilms and promotes detachment of bacterial cells from the biofilm matrix [47,48], which is advantageous for single cell analysis of bacterial cells by flow cytometry. To rule out the possibility that exposure to 0.2 mM EDTA had detrimental effects on bacterial viability, we suspended the conjugation mix in PBS containing 0.2 mM EDTA. The numbers of viable donor and recipient cells were counted before and after 1 h of treatment in EDTA-containing PBS using the dilution plate count method. We did not identify differences in viable cell counts with or without EDTA treatment. During flow cytometry data analysis, non-single cell events were further excluded from total cell counts using a doublet discrimination procedure described in Appendix A (Appendix A).

Single-color positive control strains of R and D were also incubated under the same experimental conditions and sampled every 2 h. The objectives were to verify that fluorescence emissions from the donor and recipient strains were stable over time and to allow a universal gate to be drawn on the red vs. blue fluorescence bivariate plots to distinguish these single-color positive controls from the non-fluorescent J53Az negative control strain during data analysis. The proportions of recipient (R), donors (D) and transconjugants (T) in the recovered conjugation mix were calculated for every time point (Figure 4). This allowed the transconjugant frequency T/(T + R) to be characterized for 6 h of filter mating (Figure 5A).

To validate these flow cytometry measurements, we repeated the same filter mating protocol but employed the conventional selective agar plate count method. Transconjugant frequencies were measured using T/(T + R) from colony counts (Figure 5B). A comparison between the two methods (Figure 5A,B) clearly showed that the two methods generated broadly comparable results. After 2 h of mating, the transconjugant frequencies were nearly the same, but for later time points (4 h and 6 h), the average measurements from the flow cytometry method were higher than those from the conventional method.

During conjugation, horizontal gene transfer is not the only process that contributes to the formation of transconjugants. Clonal expansion, which represents the vertical transmission of plasmids due to cell division of transconjugants, is a confounding factor that can inflate estimates of horizontal gene transfer rate [10]. To limit the confounding effects of clonal expansion, we used stationary phases of bacterial cultures, during which bacteria generally do not divide [49,50], and limited the duration of filter mating to 6 h. An advantage of pairing mCherry with *ebfp2* instead of *gfp* is that conjugating bacteria can be stained with the SYTOX Green, which allows dead cells to be counterstained and excluded [51,52]. The emission spectra of eBFP, SYTOX Green and mCherry fluorophores are spaced sufficiently apart for them to be distinguished after spectral compensation (Appendix A, Appendix A). The densities of viable bacteria can be quantified using CytoFLEX flow cytometers, which provide fast and accurate volumetric cell counts of bacteria [53,54]. It is therefore possible to extend the experimental framework in this study to obtain measurements of exconjugant frequency (T/R_0_) and conjugation frequency (T/D_0_), for which the absolute numbers of viable recipient (R_0_) or donor cells (D_0_) at the start of conjugation and viable transconjugants (T) after the conjugation are required (Table 1). It has been increasingly acknowledged that a lack of robust methods for measuring bacterial conjugation is a major barrier to interpreting and comparing existing experimental results across different studies [10,55]. It is hoped that the proof-of-principle experiment presented in our study will contribute toward the development of novel methods for quantifying rates at which conjugative plasmids are transferred across bacterial strains. Tagging therapeutic interference plasmids such as pJIMK46 using the experimental pipeline presented in this work will also enhance our ability to characterize their transfer dynamics across bacterial strains in vitro and in vivo. This is expected to shed more insights into the mechanisms through which plasmid incompatibility contributes to the phenomenon of plasmid curing.

## Figures and Tables

**Figure 1 microorganisms-11-00878-f001:**
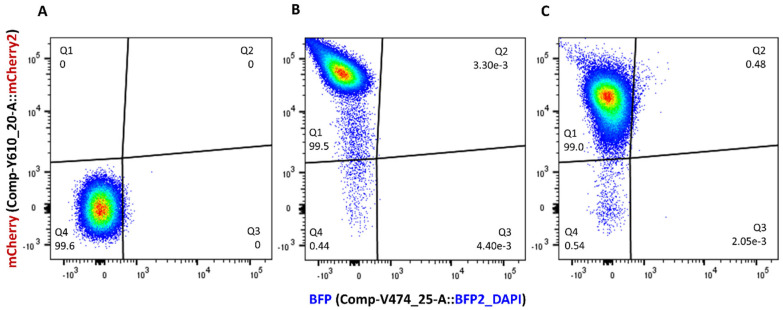
Flow cytometry detection of the *E. coli* J53Az + p_mCherry-stable recipient strain displayed on bivariate plots of red versus blue fluorescence with spectral compensation. In contrast to the non-fluorescent J53Az negative control strain (**A**), the J53Az + p_mCherry-stable single-color, positive control strain recovered from filter mating incubation conditions at time points 0 h (**B**) and 6 h (**C**) emitted strong red fluorescence signals that clearly distinguished them from the negative control strain. Regions Q1 and Q4 on the red (mCherry2) versus blue (BFP) fluorescence bivariate plots were designated mCherry^+^ and mCherry^−^ populations, respectively.

**Figure 2 microorganisms-11-00878-f002:**
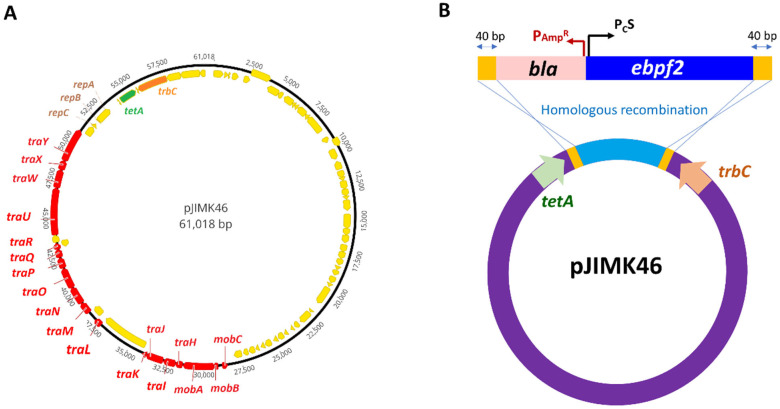
Blue fluorescent protein gene-tagging of low-copy-number IncM plasmid pJIMK46 via homologous recombineering. (**A**) The *tra* genes and *mob* genes of the conjugation machinery of pJIMK46 are shown in red. The positions of the *repA*, *repB* and *repC* genes of IncM plasmids are indicated in brown. (**B**) To label pJIMK46 with a blue fluorescent protein gene (*ebfp2*), the PCR product containing the P_C_S-*ebfp2*-*bla* construct and flanked by 40 bp overhangs (in yellow), which were homologous to the designated recombineering sites, was electroporated into a lambda-red recombinase-expressing *E. coli* host that also harbors pJIMK46 (Schematic not drawn to scale).

**Figure 3 microorganisms-11-00878-f003:**
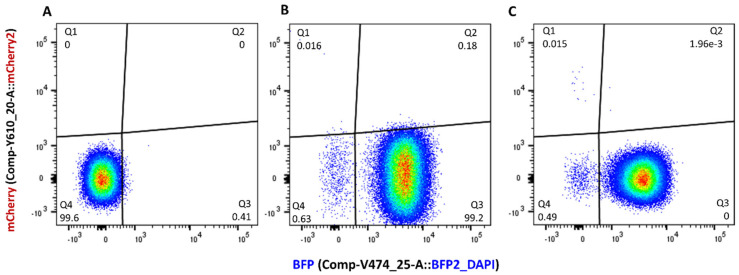
Flow cytometry detection of the *E. coli* UB5201Rf + pConj_blue-strong donor strain displayed on bivariate plots of red (mCherry) versus blue fluorescence with spectral compensation. Compared to the non-fluorescent J53Az double-negative control strain (**A**), the UB5201Rf + pConj_blue-strong single-color, positive control strain recovered from filter mating incubation conditions at time points 0 h (**B**) and 6 h (**C**) emitted strong blue fluorescence signals that distinguished them very clearly from the negative control. Regions Q1 and Q4 on the red (mCherry2) versus blue (BFP) fluorescence bivariate plots were defined as *ebfp2*^+^ and *ebfp2*^−^ populations, respectively.

**Figure 4 microorganisms-11-00878-f004:**
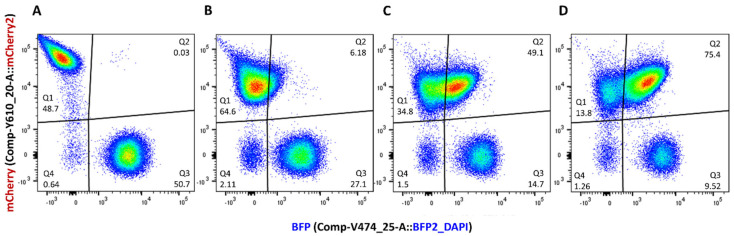
Flow cytometry results for time course filter mating between J53Az + p_mCherry-stable recipient and UB5201Rf + pConj_blue-strong donor strains over 6 h. Bacterial cells recovered from conjugation mixes at time points 0 h (**A**), 2 h (**B**), 4 h (**C**) and 6 h (**D**) were analyzed by flow cytometry. Regions Q1, Q2, and Q3 on the red (mCherry2) versus blue (BFP) fluorescence bivariate plots correspond to the recipients (mCherry^+^ *ebfp2*^−^), the transconjugants (mCherry^+^ *ebfp2*^+^) and the donors (mCherry^−^ *ebfp2*^+^) populations, respectively.

**Figure 5 microorganisms-11-00878-f005:**
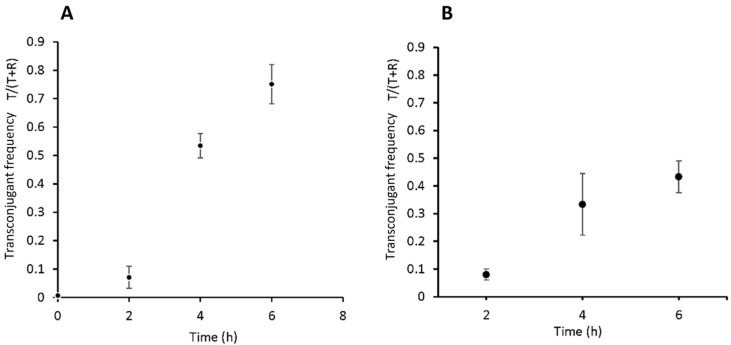
Transconjugant frequency in the conjugation mix comprising the J53Az + p_mCherry-stable recipient and UB5201Rf + pConj_blue-strong donor strains over 6 h of filter mating. Conjugation frequency measured using flow cytometry (**A**) and the conventional selective agar plate count method (**B**). Data points are the mean T/(T + R) of four biological replicates (*n* = 4) with error bars denoting standard error of the mean.

**Table 1 microorganisms-11-00878-t001:** Measured quantities in commonly used methods for characterizing frequencies at which conjugative plasmids are transferred from donors (D) to recipients (R) to form transconjugants (T) during bacterial conjugation.

Measured Quantity	Unit	Descriptions	Reference
Transconjugant frequency; T/(T + R)	Dimensionless	Ratio of the number of transconjugants to the combined populations of transconjugants and recipients at the time of measurement; also known as transfer frequency, proportion (or fraction) of transconjugants in literature	[7,18,19,20]
Transconjugant frequency; T/(DR)	mL CFU^−1^	Ratio of the number of transconjugants to the product of the number of donors multiplied by the number of recipients at the time of measurement	[21]
Exconjugant frequency; T/R_0_	Dimensionless	The number of transconjugants at the end of the conjugation per recipient cell at the start of conjugation (R_0_)	[22,23]
Conjugation efficiency; T/D_0_	Dimensionless	The number of transconjugants at the end of the conjugation per donor cell at the start of conjugation (D_0_)	[24,25]

**Table 2 microorganisms-11-00878-t002:** *E. coli* host strains used in this study.

Bacterial Strain	Characteristics	Reference
*E. coli* J53Az	A sodium azide-resistant derivative of the *E. coli* K-12 reference strain	[36]
*E. coli* UB5201Rf	Rifampicin-resistant derivative of the *E. coli* UB5201 wild-type strain with amino acid substitution S512P in the β-subunit of RNA polymerase encoded by *rpoB*	[38]

**Table 3 microorganisms-11-00878-t003:** Oligonucleotide primers used in this study.

Primer Name	Oligonucleotide Sequence (5′->3′) ^a^
mCherry_*Xba*I-F	CCA***TCTAGA***GTGAGCAAGGGCGAGGAGGAT
mCherry_*Sac*I-R	GCT***GAGCTC***ACGATTCTTTTACTTGTAC
*ccdAB*_*EC*_*Hpa*I-F	CG***GTTAAC***ACGAAACGGGAATGCGGTAA
*ccdAB*_*EC*_*Sac*I-R	GC***GAGCTC***ATGACTGCAGACTGGCTGTGT
P_c_S_*Xba*I-F	CAC***TCTAGA***AAACGGATGAAGGCACGAAC
P_c_S_*Xba*I-RBS-R	GAC***TCTAGA***TCCTCCTTTGCTGCTCCATAACATCAA
pJIMK46::P_c_S-*ebfp2-*F	**AGGCCTATGCCATGCGGGTCAAGGCGACTTCCGGCTACTT** GAGTAAACTTGGTCTGACAG
pJIMK46::P_c_S- *ebfp2-*R	**TTTGCCGGCGTTGTTAATCAGGAGGCCAAACGATGGCTGA** GGGATCCGAATTCAGGAGGT

^a^ The underline bases in the primer sequences bind to DNA templates for PCR. Bases in italics are restriction digest cut sites. Nucleotide sequences of flanking regions for homologous recombineering are shown in bold.

**Table 4 microorganisms-11-00878-t004:** Plasmids used in this study.

Plasmid	Characteristics ^b^	Source
pSS9	A pBR322-based vector containing a green fluorescent protein gene (*gfpuv*); Tc^R^	[39]
pSS9_mCherry	pSS9 with an mCherry red fluorescent protein gene and the *ccdAB_EC_* toxin–antitoxin system of an *E. coli* IncF plasmid cloned into the *Xba*I/*Sac*I and *Sac*I/*Hpa*I sites, respectively; Tc^R^	This study
pBAD33-Gm	A low–medium-copy-number vector with the p15A origin of replication and an L-arabinose inducible promoter; Gm^R^	[33,34,40]
p_mCherry-stable	pBAD33 with mCherry-*ccdAB_EC_* cloned into the *Xba*I/*Hind*III site, followed by single-site cloning of the P_c_S promoter into *Xba*I; Gm^R^	This study
pJIMK46	IncM interference plasmid derived from pJIBE401 clinical plasmid by deleting the multi-drug resistance (MDR) region and the toxin gene *pemK* of the *pemIK* TA system	[24]
pMini_blue-strong	Codon-optimized enhanced blue fluorescent protein gene (*ebfp2*) was cloned into the multiple cloning site of the NEB cloning vector pMiniT downstream of a strong constitutive promoter P_c_S; Amp^R^	This study
pConj_blue-strong	pJIMK46 was tagged with P_c_S*-ebfp2-bla* between the *trbC* and *tetA* genes in pJIMK46; Amp^R^	

^b^ Abbreviations for antibiotic resistance phenotypes include Tc^R^: tetracycline resistance; Amp^R^: ampicillin resistance and Gm^R^: gentamicin resistance.

## Data Availability

All flow cytometry raw data generated in this work were deposited in the Dryad Data Repository: https://doi.org/10.5061/dryad.12jm63z2t. The plasmid maps of key plasmids generated in this work can also be downloaded from this URL in GenBank format.

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
