# Peer review of "A Streamlined Approach for Fluorescence Labelling of Low-Copy-Number Plasmids for Determination of Conjugation Frequency by Flow Cytometry"

_microorganisms, 2023, doi:10.3390/microorganisms11040878_

Round 1

Reviewer 1 Report

This manuscript, “A streamlined approach for fluorescence labelling of low copy-number plasmids for determination of conjugation frequency by flow cytometry" by Qi and colleagues, describe a flow cytometry-based method to accurately determine the frequency of the DNA transfer, namely conjugation, between E. coli bacteria.

The quantification of DNA transfer during bacterial conjugation is a persistent challenge in the field of microbiology. To address this issue, the authors develop a method based on the high expression of two fluorescent reporter genes present in the donor cells on the plasmid to be transferred and in the recipient cells on medium copy-number plasmid. This latter small plasmid carry a toxin/anti toxin system ensuring that cell free segregants would be rapidly eliminated, thus increasing the reliability of the frequency of transfer measurement. The method is straightforward to implement, since it does not require any genetic modification of the donor or recipient chromosomes. Instead, it only relies on the introduction of these plasmids in the recipient cells. To maximize the accuracy of the measurement, the conjugation is performed on filters with near-stationary phase, concentrated cells that prevent further growth. The cytometry assays are meticulously controlled to ensure that signals are accurately and specifically detected in over 99% of cells. The results of the time-lapse experiment performed on one controlled example demonstrate the reliability and accuracy of the proposed method. This methodology offers a valuable tool for the accurate quantification of DNA transfer during bacterial conjugation, and may have broad applications in various areas of microbiology research.

The manuscript is well written and the data support the conclusion. However, there are two minor concerns that should be addressed to further enhance the overall quality of the manuscript.

Minor comments:

- The authors should compare and discuss their results with the frequency of conjugation observed with at least one of the other methods.

- They should also clearly indicate and discuss the range of frequency transfer for which this method is accurate.

Other comments:

- L. 318: "this plasmid has a low copy-number p15A replicon". Please change to medium copy-number (12-15 copies per chromosome) and indicate a reference.

- Figures: Please remove the "Q1 / number" labelling in the cytometry figures that overlaps the actual data.

Author Response

We sincerely thank the reviewer-1 for his valuable comments and constructive feedback, which helped us improve our manuscript.

Reviewer-1 

The manuscript is well written and the data support the conclusion. However, there are two minor concerns that should be addressed to further enhance the overall quality of the manuscript.

Minor comments:

  • The authors should compare and discuss their results with the frequency of conjugation observed with at least one of the other methods.

Response: We now measured conjugation frequency using the conventional selective agar plate counting method (Fig. 5B) and compared the results with flow cytometry-based measurement in the methods (lines 200-209) and results (lines 347-353) sections in the revised manuscript. The results obtained using both experimental techniques were broadly comparable.

  • They should also clearly indicate and discuss the range of frequency transfer for which this method is accurate.

Response: This flow cytometry method is expected to be accurate as long as two single-color positive control strains can be clearly distinguished from the double-negative, non-fluorescent control strain via gating at all the time points during which measurements were taken. For this reason, we suggested in our revised manuscript that users should ensure that “single color controls such as those shown in Figure 1 are included throughout the time course of the conjugation assays” (lines 261-263).

Other comments:

  • 318: "this plasmid has a low copy-number p15A replicon". Please change to medium copy-number (12-15 copies per chromosome) and indicate a reference.

Response: Although p15A has been described as “low-copy number” and “medium copy-number” replicons in literature, we believe that it is a low-medium copy-number replicon, because its copy number is lower compared to a typical medium copy-number replicon such as pBR322 ori (~20). We have replaced “low” with “low-medium” when referring to plasmid backbones with p15A ori throughout the manuscript and in Table 4. We also included citations to Kim et al (2015) and Chang & Cohen (1978).

  • Figures: Please remove the "Q1 / number" labelling in the cytometry figures that overlaps the actual data.

Response: We thank the reviewer for this excellent point which we have taken on board. All the flow cytometry figures have been updated accordingly. We also uploaded our flow cytometry raw data to the Dryad Data Repository.

Reviewer 2 Report

This manuscript describes a workflow to create and analyze conjugation efficiency in low copy number plasmids. The methodology utilizes a) a stable plasmid encoded mCherry reporter in the recipient b) lambda recombineering to create a bfp conjugative plasmid in the donor and c) flow cytometry to analyze the data in the absence of antibiotic selection. Although most of the procedures described here have been previously published, the authors combined them into an easy-to-follow workflow and have some key advantages, namely the high expression promoter used to get high fluorescent protein expression from a low copy plasmid and the newly created mCherry plasmid which is stabilized by a toxin-antitoxin system and thus is easy to maintain.

1) I think it is essential that the authors compare the results of their mating protocol and flow cytometry with more classic antibiotic selection. They are arguing that this method is better because it does not require antibiotic selection. I certainly can see that as an advantage, but I would expect that under control conditions as used here (ie, no additional stress) that there is likely a good correlation between the flow cytometry measurements and plating on antibiotics (Gm and azide or Tet).  And if not, this should be discussed.

2) This manuscript would be much easier to read if shorter names for the plasmids and strains were used in the text (e.g., give another name to J53Az + pBAD33::PCS-mCherry-ccdAB)

3) The axis labelling in Fig 1, 3 and 4 is confusing. E.g., I don’t know what is meant by ‘Comp-V474_25-A::BFP2_DAPI’. If it is needed it should be defined and only be used in the legend; the axis labelled blue (or red).

4) The authors state that they used stationary phase cells and only 6 h of growth in the mating to minimize cell growth during the experiment. I understand why they did this but I doubt very much that the bacteria do not grow in 6 hours on an LB plate. Further, there are reports of growth phase variation in conjugation efficiency. Can the authors suggest a way to correct for this growth?

5) EDTA was added to the diluted cells. Have the authors confirmed (either in the literature or in their lab) that this has no effect on the viability of either the donor or recipient?

6) It would be nice if the template for making the bfp plasmids and the mCherry recipient plasmid were deposited in Addgene to make it more easily accessible.

Author Response

We sincerely thank  reviewer-2 for his valuable comments and constructive feedback, which helped us improve our manuscript.

  • I think it is essential that the authors compare the results of their mating protocol and flow cytometry with more classic antibiotic selection. They are arguing that this method is better because it does not require antibiotic selection. I certainly can see that as an advantage, but I would expect that under control conditions as used here (ie, no additional stress) that there is likely a good correlation between the flow cytometry measurements and plating on antibiotics (Gm and azide or Tet).  And if not, this should be discussed.

Response: We now measured conjugation frequency using the conventional selective agar plate counting method (Fig. 5B) and compared the results with flow cytometry-based measurement in the methods (lines 200-209) and results (lines 347-353) sections in the revised manuscript. The results obtained using both experimental techniques were broadly comparable.

  • This manuscript would be much easier to read if shorter names for the plasmids and strains were used in the text (e.g., give another name to J53Az + pBAD33::PCS-mCherry-ccdAB)

Response: We have replaced the longer plasmid and strain names with shorter versions throughout the manuscript and in Table 4.

  • The axis labelling in Fig 1, 3 and 4 is confusing. E.g., I don’t know what is meant by ‘Comp-V474_25-A::BFP2_DAPI’. If it is needed it should be defined and only be used in the legend; the axis labelled blue (or red).

Response. These full-axis labels were generated by the BD Symphony flow cytometer. This labeling convention can be seen in flow cytometry literature. For example, “Comp” indicates that compensation was applied to correct for inherent fluorescence spillovers between detectors. “V474_25” refers to the laser used (violet in this case), wavelength information etc. We agree with the Reviewer that this could be confusing for readers who are not familiar with this labeling system and have updated the axes in Figures 1,3 and 4 with both abbreviated labeling and full labelling, i.e. BFP (Comp-V474_25-A::BFP2_DAPI) and mCherry (Comp-Y610_20-A::mCherry2), in the revised manuscript.

  • The authors state that they used stationary phase cells and only 6 h of growth in the mating to minimize cell growth during the experiment. I understand why they did this but I doubt very much that the bacteria do not grow in 6 hours on an LB plate. Further, there are reports of growth phase variation in conjugation efficiency. Can the authors suggest a way to correct for this growth?

Response: We agree with the reviewer that bacterial numbers are expected to increase even during the stationary phase. The quantity transconjugant frequency (T/T+R) used in this manuscript corrects for such inevitable increase or decrease of bacterial cells over time, because T+R represents all recipients at a specific time.

  • EDTA was added to the diluted cells. Have the authors confirmed (either in the literature or in their lab) that this has no effect on the viability of either the donor or recipient?

Response: We suspended bacterial cells in PBS containing a low concentration of EDTA to prevent cells from clumping and forming biofilms. This ensures more accurate single cell counts and prevents clogging of flow cytometer nozzle. The suspension takes place for a maximum of 60 min and is not expected to have adverse effects on bacterial viability. To confirm this, we measured viable cell counts before and after the treatment with PBS + 0.2 mM EDTA for an hour. We did not observe any differences with the donor and recipient counts. We added description of this supplementary experiment to lines 329-335 in the revised manuscript.

2.6.  It would be nice if the template for making the bfp plasmids and the mCherry recipient plasmid were deposited in Addgene to make it more easily accessible.

Response: We thank the reviewer for this excellent idea and plan to submit our plasmids to AddGene soon. In the meantime, we have deposited the plasmid maps of the plasmids generated in this work in GenBank format to Dryad. Detailed information including allocated AddGene catalogue numbers will be included on our Dryad Data Repository page that accompanies this manuscript.